# Stochastic Ensemble Climate Forecast with an Analogue Model

Pascal Yiou[1] and Céline Déandréis[2]

[1]Laboratoire des Sciences du Climat et de l'Environnement, UMR 8212 CEA-CNRS-UVSQ, IPSL and Université Paris-Saclay, CE l'Orme des Merisiers, 91191 Gif-sur-Yvette, France
[2]ARIA Technologies, 8-10 Rue de la Ferme, 92100 Boulogne-Billancourt, France

**Correspondence:** P. Yiou (pascal.yiou@lsce.ipsl.fr)

**Abstract.** This paper presents a system to perform large ensembles climate stochastic forecasts. The system is based on random analogue sampling of sea-level pressure data from the NCEP reanalysis. It is tested to forecast a North Atlantic Oscillation (NAO) index and the daily average temperature in five European stations. We simulated 100 member ensembles of averages over lead times from 5 days to 80 days in a hindcast mode, i.e. from a meteorological to a seasonal forecast. We tested the hindcast simulations with usual forecast skill scores (CRPS or correlation), against persistence and climatology. We find significantly positive skill scores for all time scales. Although this model cannot outperform numerical weather prediction, it presents an interesting benchmark that could complement climatology or persistence forecast.

*Copyright statement.* TEXT

## 1 Introduction

Stochastic weather generators (SWG) have been devised to simulate many and long sequences of climate variables that yield realistic statistical properties (Semenov and Barrow, 1997). Their main practical use has been to investigate the probability distribution of local variables such as precipitation, temperature or wind speed, and their impacts on agriculture (Carter, 1996; Semenov, 2006), energy (Parey et al., 2014) or ecosystems (Maraun et al., 2010). Such systems can simulate hundreds or thousands of trajectories on desktop computers and propose cheap alternatives to climate model simulations.

There are many categories of SWGs (Ailliot et al., 2015). Some SWGs are explicit random processes, whose parameters are obtained from observations of the variable to be simulated (Parey et al., 2014). Some SWGs are based on a random resampling of the observations (Iizumi et al., 2012). Some SWGs simulate local variables from their dependence to large-scale variables such as the atmospheric circulation (Kreienkamp et al., 2013). This allows to simulate spatially coherent multivariate fields (Yiou, 2014; Sparks et al., 2018) and can be used for downscaling (Wilks, 1999).

SWGs that use observations as input could in principle be used to forecast variables. This is the case for analogue weather generators (Yiou, 2014). Methods of analogues of atmospheric circulation were first devised for weather forecast (Lorenz, 1969; van den Dool, 1989). They were abandoned when numerical weather prediction was developed and implemented, because their performance was deemed inadequate (van den Dool, 2007). However, recent studies on *nowcasting* have shown

that analogue based methods could outperform numerical weather prediction for precipitation (Atencia and Zawadzki, 2015). Yiou (2014) showed some skill for temperature simulations in Europe of an analogue SWG.

Due to uncertainties in observations and the high sensitivity to initial conditions (van den Dool, 2007) weather forecasts estimate probability density functions rather than deterministic meteorological values. Therefore, weather forecasts examine the properties of all possible trajectories of an atmospheric system from an ensemble of initial conditions. Such properties include the range and the median, for example. Then one can compare how the ensemble of trajectories compares to observations, and other reference forecasts. Numerical weather forecasts rely on large ensembles of model simulations and require a massive use of supercomputers in order to provide estimates of the probability density function (pdf) of variables of interest, for various lead times. Being able to increase the ensemble size of weather forecast systems in order to lower the bias of the forecast skill has been a challenge of major centers of weather prediction (Weisheimer and Palmer, 2014).

The most trivial prediction systems are based on either climatology (i.e. predicting from the seasonal average) or persistence (i.e. predicting from the past observed values) (Wilks, 1995). Probabilistic and statistical models can provide more sophisticated benchmarks for weather forecast systems, still without simulating the underlying primitive hydrodynamic equations and using supercomputers. For example, statistical models of forecast for precipitation based on analogues (of precipitation) were tested for North America (Atencia and Zawadzki, 2015). Such systems tend to outperform numerical weather forecast systems, although their computing cost is steeper than most SWGs. Therefore the potential of analogue based methods can be useful to assess probability distributions, rather than a purely deterministic forecast.

Machine learning algorithms were recently devised to simulate complex systems (Pathak et al., 2018a) with surprising performances. Such algorithms are sophisticated ways of computing analogues of observed trajectories in a learning step, and simulating potentially new trajectories from this learning. The main drawback is that such algorithms generally require a tricky tuning of parameters that might not be based on a physical intuition. From the inspiration of machine learning algorithms, we propose to devise a weather forecast system based on a stochastic weather generator that uses analogues of circulation to generate large ensembles of trajectories. The rationale for using analogues, rather than more sophisticated machine learning, is that they correspond to a physical interpretation of relations between large scale and regional scales. Moreover, mathematical results in dynamical system theory (Freitas et al., 2016; Lucarini et al., 2016) suggest that properties of recurring patterns is asymptotically independent on the distance that is used to compute analogues.

This paper presents tests of such a system to forecast temperatures in Europe and an index of the North Atlantic Oscillation (NAO). The NAO controls the strength and direction of westerly winds and location of storm tracks across the North Atlantic in the winter (Hurrell et al., 2003). Positive values of the index indicate a strengthened Azores anticyclone and a weaker Icelandic low. Negative values indicate a weak Azores anticyclone and a strong Icelandic low. The North Atlantic Oscillation is strongly tied to temperature and precipitation variations in Europe (Slonosky and Yiou, 2001).

Since the set up of such a system is fairly light, it is possible to test it for time leads from a meteorological forecast (5 days ahead) to a seasonal forecast (80 days ahead). We test this system in hindcast experiments to forecast climate variables between 1970 and 2010. The tests are performed with usual skill scores (continuous rank probability score and correlation).

The paper is organized as follows. Section 2 presents the datasets that are used as input of the system. Section 3 presents the forecast system based on analogues, the skill scores and the experimental protocol. Section 4 presents the results on simulations of the NAO index and European temperatures.

## 2    Data

5  We used data from different sources for sea-level pressure (SLP), NAO index and temperatures. SLP data are used for analogue computations as predictor. The NAO index and temperatures are the predictands (i.e. variables to be predicted). It is important that they share a common chronology, in order to allow their simulation because the NAO index and temperatures are simulated from from SLP analogues.

### 2.1    Sea-level pressure

10  We use the reanalysis data of the National Centers for Environmental Prediction (NCEP) (Kistler et al., 2001). We consider the sea-level pressure (SLP) over the North Atlantic region. We used SLP daily averages between January 1st 1948 and April 30th 2018. The horizontal resolution is 2.5° in longitude and latitude. The rationale of using this reanalysis is that it covers more than 60 years and is regularly updated, which makes it a good candidate for a continuous time forecast exercise.

One of the caveats of this reanalysis dataset is the lack of homogeneity of assimilated data, in particular before the satellite 15  era. This can lead to breaks in pressure related variables, although such breaks are mostly detected in the southern hemisphere and the Arctic regions (Sturaro, 2003). We are not interested in the evaluation of SLP trends, therefore breaks should only marginally impact our results.

### 2.2    NAO index

The North Atlantic Oscillation (NAO) is a major mode of atmospheric variability in the North Atlantic (Hurrell et al., 2003). Its 20  intensity is determined by an index that can be computed as the normalized sea-level pressure difference between the Azores and Iceland (Hurrell, 1995). The NAO index is related to the strength and direction of the westerlies, so that high values correspond to zonal flows across the North Atlantic region, stormy conditions and rather high temperatures in Western Europe (Slonosky and Yiou, 2001; Hurrell et al., 2003).

We retrieved the daily NAO index from the NOAA web site:
25  http://www.cpc.ncep.noaa.gov/products/precip/CWlink/pna/nao.shtml.

The procedure to calculate the daily NAO teleconnection indice is detailed on the NOAA web site. In short, a Rotated Principal Component Analysis (RPCA) is applied to monthly averages of geopotential height at 500 hPa (Z500) anomalies (Barnston and Livezey, 1987) in the 20N–90N region, between January 1950 and December 2000, from the NCEP reanalysis. The empirical orthogonal functions (EOFs) provide climatological monthly teleconnection patterns (Wilks, 1995). Those 30  monthly teleconnection patterns are interpolated for every day in the year. Then daily Z500 anomaly fields are projected onto the interpolated climatological teleconnection patterns in order to obtain a daily NAO index.

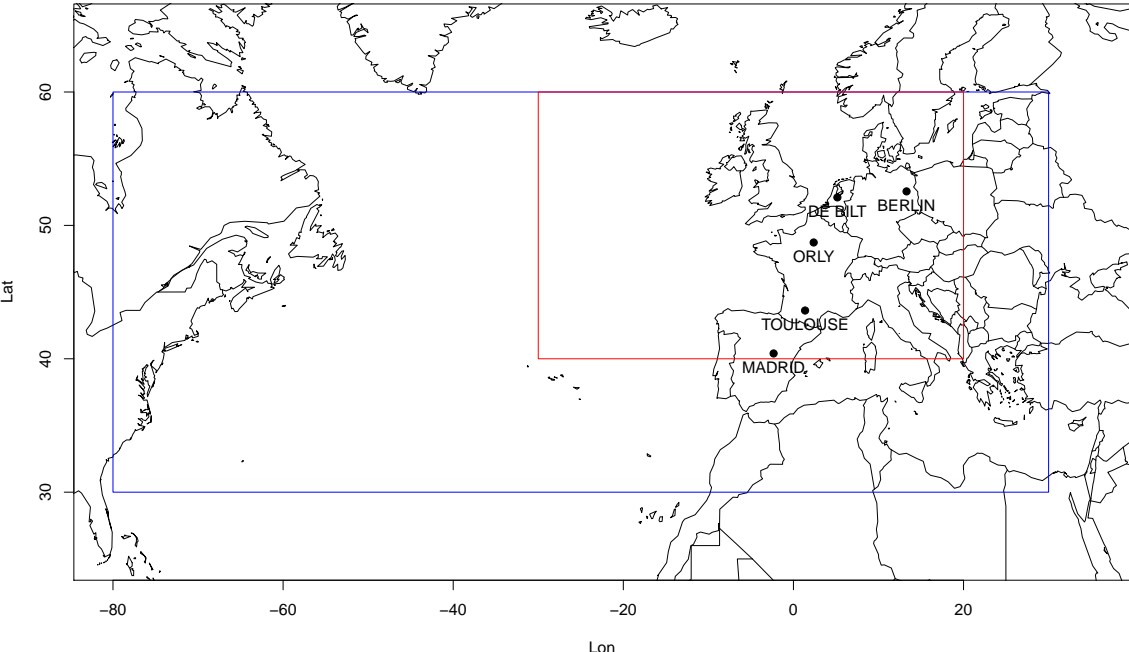

**Figure 1.** Upper panel: North Atlantic region (blue rectangle) and Western European region (red rectangle) on which analogues are computed.

The geographical domain on which this NAO index is computed is larger than the one for SLP data. Scaife et al. (2014) used an NAO index to test the UKMO seasonal forecast system. The index they used is based on monthly SLP differences between the Azores and Iceland, and is therefore different from ours.

### 2.3 European temperatures

5  We took daily averages of temperatures from the ECAD project (Klein-Tank et al., 2002). We extracted data from Berlin, De Bilt, Toulouse, Orly and Madrid (Fig. 1). Those five stations cover a large longitudinal and latitudinal range in western Europe. These datasets were also chosen because

– they start before 1948 and end after 2010. This allows the computation of analogue temperatures with the SLP from the NCEP reanalysis, which includes that period,

10  – they contain less than 10% of missing data.

These two criteria allow keeping 528 out of the 11422 ECAD stations that are available in 2018.

## 3 Methods

### 3.1 Analogues of circulation

Analogues of circulation are computed on SLP data from NCEP (Sec. 2.1). For each day between Jan. 1st 1948 and Dec. 31st 2017, the best 20 analogues (with respect to a Euclidean distance) in a different year are searched. This follows the procedure of (Yiou et al., 2013). The analogues are computed over two regions (large region: North Atlantic region (80W–30E; 30–70N); small region: Western Europe (30W–20E; 40–60N)). The large region is used to simulate/forecast the NAO index. This choice is justified by the fact that the North Atlantic atmospheric circulation patterns are well defined over that region (Michelangeli et al., 1995). The small region is used to simulate/forecast continental temperatures, following the domain recommendations of the analysis of Jézéquel et al. (2018).

### 3.2 Forecast with analogue stochastic weather generator

Ensembles of simulations of temperature or the NAO index can be performed with the rules illustrated by Yiou (2014), with an analogue-based stochastic weather generator. This stochastic weather generator can be run in so called *dynamic* mode. For each initial day $t^{(1)}$, we have $N$ best SLP analogues. We randomly select one ($k$) of those $N$ analogues and time $t_k^{(1)}$, with a probability weight that is

1. inversely proportional to the calendar distance of the analogues dates $t_k^{(1)}$ to $t^{(1)}$. This constrains the time of analogues to move forward.

2. inversely proportional to the correlation of the analogue with the SLP pattern at time $t^{(1)}$. This constraint favors analogues with the best patterns, among those with the closest distance.

3. a zero weight if $t_k^{(1)}$ is larger than $t^{(1)}$. This ensures that no information coming from times beyond $t^{(1)}$ is used in the simulation process.

The simulated SLP at the next day $t^{(2)}$ is then the next day of the selected analogue ($t^{(2)} = t_k^{(1)} + 1$). We repeat this operation on $t^{(2)}, \dots t^{(t)}$ until a lead time $T$. This generates one random daily trajectory between $t^{(1)}$ and $t^{(1)} + T$. The random sampling procedure is repeated $S$ times to generate an ensemble of trajectories. Here, $S = 100$. This procedure is summarized in Fig. 2.

If we want to simulate a daily sequence starting at time $t$ and until $t + T$, we have excluded all analogues whose date falls in $[t, t+T]$ in the random analogue selection. This provides a simple way of performing hindcast forecast for temperature or NAO index.

In this paper, the lead time $T$ is 5, 10, 20, 40 and 80 days ahead. The latter two values are meant to illustrate the limits of the system. For each daily trajectory starting at $t$, we compute the temporal average between $t$ and $t + T$. Therefore, we go from an ensemble meteorological forecast (5 days) to a seasonal forecast (80 days) of averaged trajectories.

The $S = 100$ simulations at each time steps allow computing medians and quantiles of the averaged trajectories.

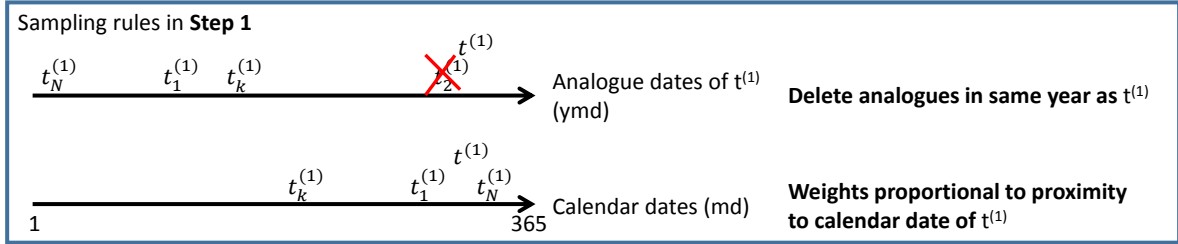

**Figure 2.** Schematic of the iteration procedure to simulate one random trajectory of temperature (TG) from SLP analogues. The values of $t^{(k)}$ are the days to be simulated by the system. The values of $t_1^{(k)}, \ldots, t_N^{(k)}$ are the analogue days for $t^{(k)}$. The red SLP rectangles are the randomly selected analogues, according to the rule defined in the lower box. This procedure is repeated $S$ times to generate an ensemble of trajectories.

For comparison purposes, climatological and persistence forecasts are also computed. The climatological forecast for a lead time $T$ is determined from the seasonal cycle of $T$ averages of the variable we simulate. For each time $t$, the *climatological* forecast for $t + T$ is the mean seasonal cycle of $T$ averages at the calendar day of $t$. The *persistence* forecast at time $t$ for a lead time of $T$ is the observed average between $t - T$ and $t$. Those two types of forecasts are illustrated in Fig. 3. Ensembles of

5  reference forecasts are performed by adding a Gaussian random noise (independent and identically distributed), whose variance is the variance of the observed $T$ averages. These two definitions ensure a coherence between the predictand (averages over $T$ values ahead) and predictors for references (mean of averages over $T$ values for climatology, or average over $T$ preceding values for persistence).

### 3.3 Alternative autoregressive weather generator

10  This non parametric weather generator (based on data resampling) is compared to a parametric autoregressive simple model, based on a similar principle of a relation between SLP and variables like temperature and NAO index. We build a multi-variate

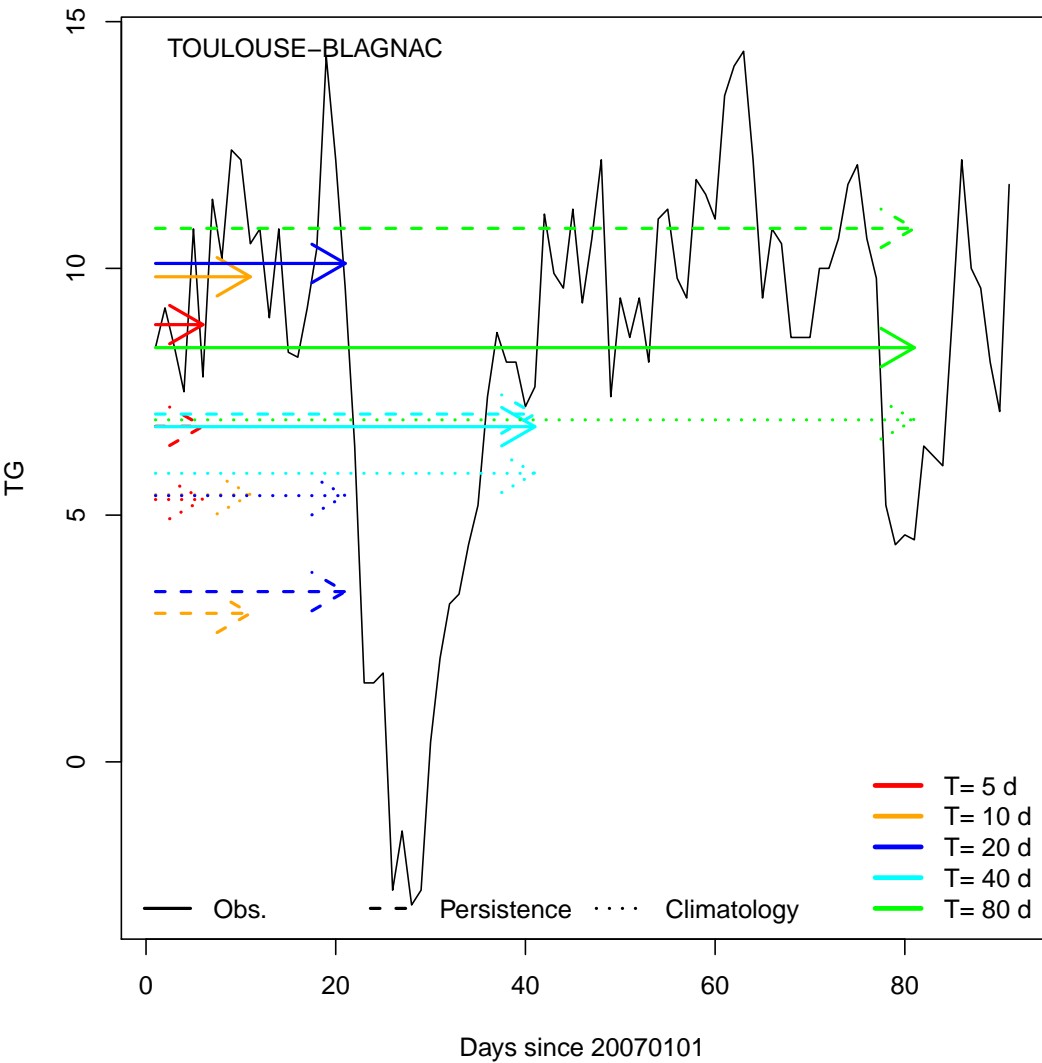

**Figure 3.** Illustration of average forecast for daily mean temperature (TG) in Toulouse, for Jan. 1st 2007. The continuous black line indicates the observations of TG for the first 90 days of 2007. The colors indicate lead times $T$. The continuous arrows are for averages of observed TG from Jan. 1st 2007 to the lead time $T$. The dashed lines are for the persistence forecast of TG and the dotted lines are for the climatology forecast of TG on Jan. 1st 2007.

autoregressive model of order 1 $R_t$ for SLP by expressing:

$$R_{t+1} = A \cdot R_t + B_t, \tag{1}$$

where $A$ is a "memory" matrix and $B_t$ is multivariate random Gaussian noise with covariance matrix $\Sigma$. We impose that the multivariate process $R_t$ yields the same covariance matrix $C(0)$ and same lag-1 covariance matrix $C(1)$ as SLP. The matrices $A$ and $\Sigma$ can be estimated by:

$$A = C(1)^t \cdot C(0)^{-1} \tag{2}$$

and

$$\Sigma = C(0) - C(1)^t \cdot A. \tag{3}$$

The superscript $^t$ is matrix transposition. In order to avoid numerical problems in the estimation of $C(0)^{-1}$, the model in Eq. (1) is formulated on the first 10 principal components (von Storch and Zwiers, 2001) of North Atlantic SLP (80W–30E; 30–70N: blue rectangle in Figure 1), which account for $\approx 80\,\%$ of the variance. In this way, $C(0)$ is a diagonal matrix whose elements are the variances of the 10 principal components of SLP. Such a parametric model has been used as a null hypothesis for weather regime decomposition by Michelangeli et al. (1995).

We then perform a multilinear linear regression between the five mean daily temperature records (TG at Berlin, Toulouse, Orly, Madrid, De Bilt) and the NAO index:

$$X_t = a\mathrm{SLP}_t + b + \epsilon_t \tag{4}$$

where $X = (\mathrm{TG}_{\mathrm{Berlin}}, \ldots, \mathrm{TG}_{\mathrm{DeBilt}}, \mathrm{NAO})$, $a$ is a $6 \times 10$ matrix, $b$ is a 10-dimensional vector, and $\epsilon_t$ is a 10 dimensional residual term. We simulate Eq. (1) with the same observed initial conditions as for the analogue forecast. Then Eq. (4) is applied to simulate an ensemble of forecasts of temperatures and NAO index. In this multivariate autoregressive model (mAR1), the temporal atmospheric dynamics is contained in the matrix $A$. The major caveat of the parametric model in Eqs. (1 and 4) is that it does not contain a seasonal cycle. Introducing a seasonal dependence on the matrices $A$ and $a$ would require many tests that are beyond the scope of this paper.

### 3.4 Forecast skill

The simplest score we use is the temporal correlation between the median of the ensemble forecast and the observations. Due to the autocorrelation and seasonality of the variables we try to simulate (temperature and NAO index), we consider the correlations for the forecast in the months of January and July.

The continuous rank probability score (CRPS) compares the cumulated density functions of a forecast ensemble and observations $y_t$, for all times $t$ (Ferro, 2014).

$$\mathrm{CRPS}(t) = \int_{-\infty}^{\infty} \left( F_t(x) - \mathbf{1}(x \geq y_t) \right)^2 dx. \tag{5}$$

$F_t$ is the cumulated density function of the ensemble forecast at time $t$. It is obtained empirically from the ensemble of simulations of the model. $\mathbf{1}(x \geq y_t)$ is the empirical cumulated density function of the observation $y_t$.

The score is the average over all times:

$$\text{CRPS} = \frac{1}{N} \sum_{t=1}^{N} \text{CRPS}(t). \tag{6}$$

The CRPS is a fair score (Ferro, 2014; Zamo and Naveau, 2018) in that it compares the probability distributions of forecasts and observations and it is optimal when they are the same. Discrete estimates of CRPS can yield a bias for small ensemble sizes $S$. We simulate $S = 100$ trajectories for each forecast. This is more than most ensemble weather forecasts (typically, $S = 51$ for the European Center for Medium Range Forecast (ECMWF) ensemble forecast) and guaranties that the bias due to the number of samples is negligible. A perfect forecast gives a CRPS value of 0.

The CRPS can be decomposed into a reliability, resolution and uncertainty terms (Hersbach, 2000, Eq. (35)):

$$\text{CRPS} = \text{Reli} - \text{Resol} + \text{U}. \tag{7}$$

The reliability $\text{Reli}$ term measures whether events that are forecast with a certain probability $p$ did occur with the same fraction $p$ from the observations (Hersbach, 2000). The remaining terms of the right hand side of Eq. (7) are called the potential CRPS, i.e. it is the CRPS value one would obtain if the forecast were perfectly reliable ($\text{Reli} = 0$). Hersbach (2000) argues that the

15 potential CRPS is sensitive to the average spread of the ensemble.

     The units of CRPS are those of the variable to be forecast, therefore its interpretation is not universal, and comparing the CRPS values for NAO index and temperatures is not directly possible. Therefore, it is useful to compare the CRPS of the forecast with the one of a reference forecast. A normalization of CRPS provides a skill score with respect to that reference:

$$\text{CRPSS}_{\text{ref}} = 1 - \frac{\text{CRPS}}{\text{CRPS}_{\text{ref}}}. \tag{8}$$

The CRPSS indicates an improvement over the reference forecast. A perfect forecast has a CRPSS of 1. A positive improvement over the reference yields positive a CRPSS value. A value of 0 or less indicates that the forecast is worse than the reference.

     We compute CRPSS for the climatological and persistence references. We used the packages "SpecsVerification" and "verification" in R to compute CRPS decomposition and CRPSS scores. Hence we compare our stochastic forecasts with forecasts

made from climatology and persistence. By construction, the persistence forecast shows an offset with the actual value ahead, because the persistence is the value of the average of observations between $t - T$ and $t$. The variability of the climatological forecast is low because it is an average of $T$ long sequences.

### 3.5 Protocole

We tested the ensemble forecast system on the period between 1970 and 2010. We simulate $N = 100$ trajectories of lengths $T \in \{5, 10, 20, 40, 80\}$ days for a given date $t$, and average each trajectory over $T$. The dates $t$ are shifted every $\delta t \in \{2, 5, 10, 10, 20\}$ days, respectively for each different value of lead times $T$.

We recall that the tests we perform are on the *average* of the forecast between $t$ and $t + T$, not on the *value at* time $t + T$.

The CRPS and CRPSS is computed for each value of lead times $T$, with references of climatology and persistence. We determine the CRPS reliability and plot quantile-quantile plots for observed and forecast values of the averages. This allows assessing biases in simulating averages. Variables such as temperature yield a strong seasonality, which is larger than daily variations. It is hence natural to have very high correlations or skill scores if one considers those scores over the whole year. Therefore we compare the skill scores for January and July, in order to avoid obtaining artificially high scores.

## 4 Results

We performed our stochastic forecasts on the NAO index and European temperatures with the analogue stochastic weather generator and the mAR1 model. The two datasets (NAO and temperature) are treated separately because the simulations are done with two different analogue computations (Sec. 3.1).

### 4.1 NAO index

For illustration purposes, we comment on the skill on simulations of 2007. Fig. 4 shows the simulated and observed values of averages of the NAO index, for five values of $T$ (5, 10, 20, 40 and 80 days). This example suggests a good skill to forecast the NAO index from SLP analogues, especially at lead times of $T = 5$ to 10 days.

The q-q plots of the median of simulations versus observations show a bias that reduces the range of variations (Fig. 4, right column). There are two reasons for this reduction of variance, which is proportional to the lead time $T$:

1. individual simulated trajectories tend to "collapse" toward a climatological value after $\approx 10$ days,

2. taking the median of all simulations also naturally reduces the variance.

The q-q plots are almost linear. This means that the bias could in principle be corrected by a linear regression. We will not perform such a correction in the sequel.

The correlation, CRPS reliability and CRPSS values for NAO index forecast are shown in Fig. 5. The values of $\mathrm{CRPSS_{pers}}$ (for a persistence reference) are rather stable (with a slight increase) near 0.45, and the climatology score slightly decreases with $T$ although positive.

The CRPS reliability values range from $5 \cdot 10^{-3}$ (10 days) to 0.01 (40 days). If they are normalized to the CRPS value (or the variance of the NAO index), this is in the same range as the results of Hersbach (2000) for the ECMWF forecast system up to 10 days.

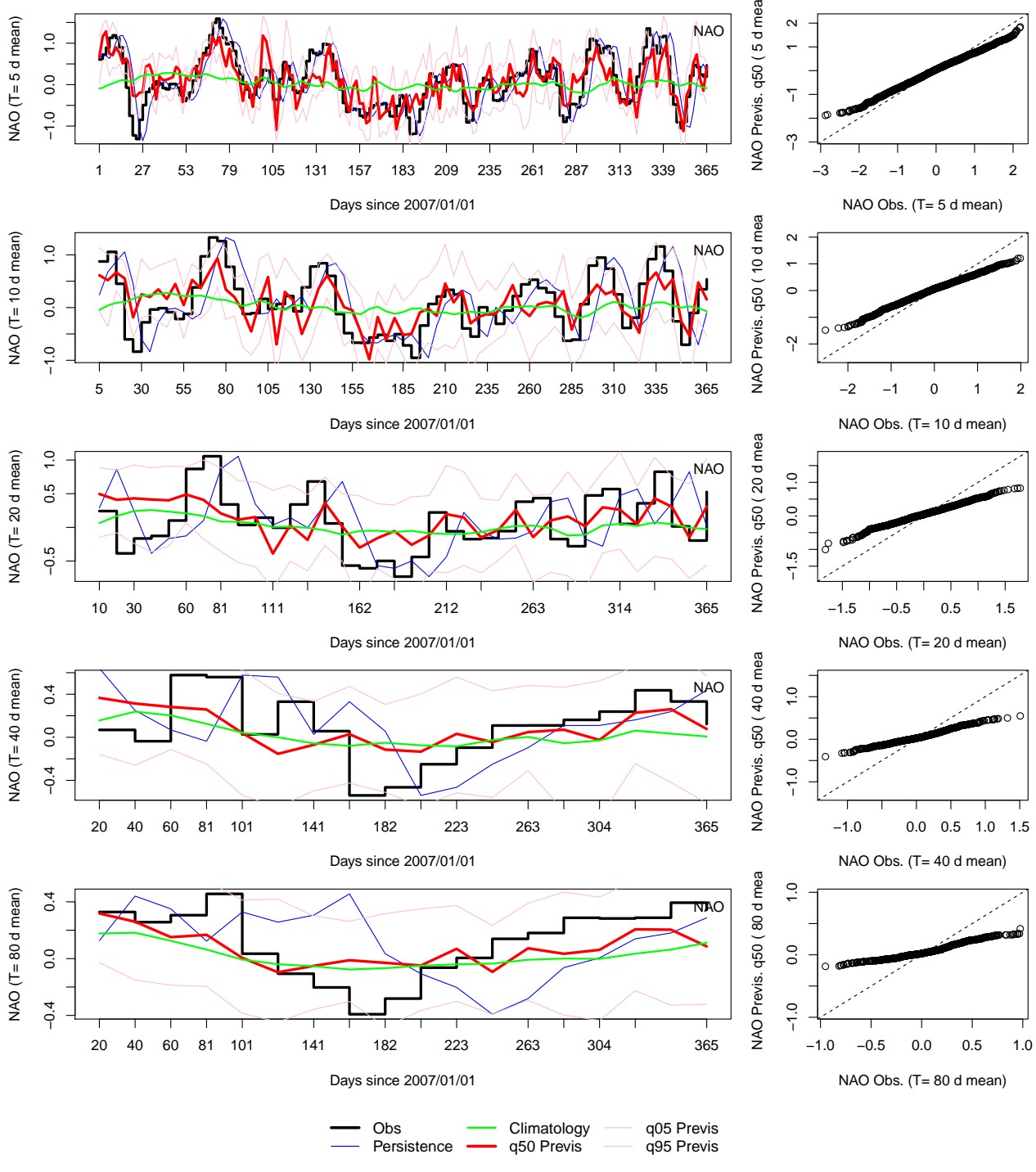

**Figure 4.** Left column: time series of analogue ensemble forecasts for 2007, for lead times $T \in \{5, 10, 20, 40, 80\}$ days. Red lines represent the median of 100 simulations; pink lines represent the 5th and 95th quantiles of the 100 member ensemble. Right column: q-q plots of NAO forecasts vs. observed values for all years in 1970–2010 for all lead times. The dotted line is the first diagonal.

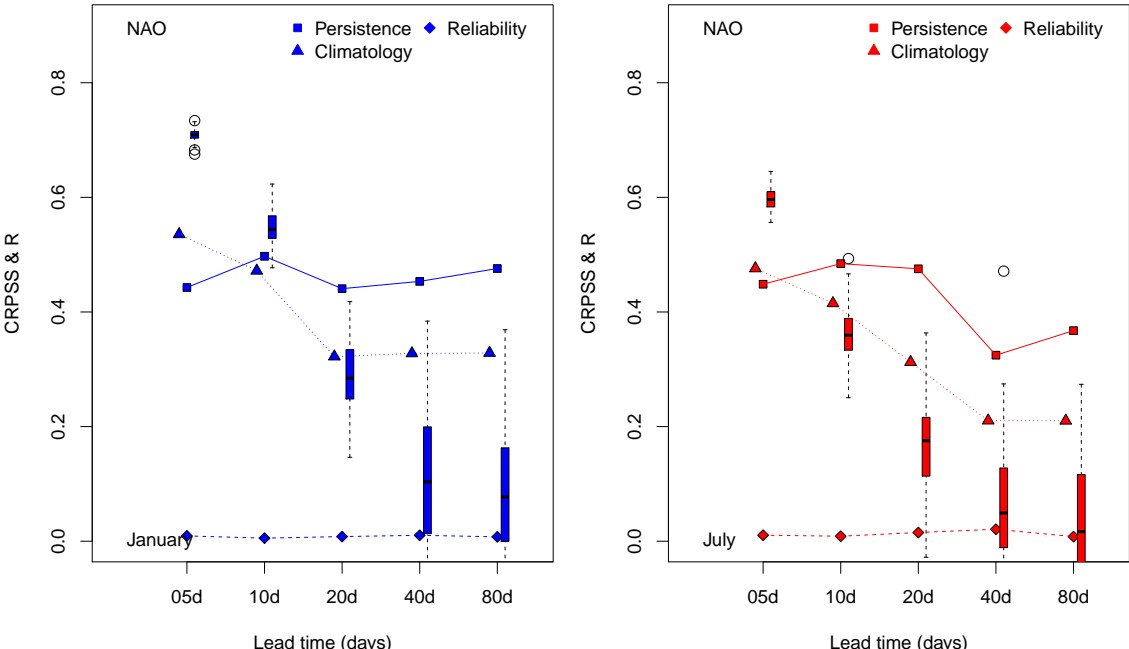

**Figure 5.** Skill scores for NAO index for lead times $T$ of 5, 10, 20, 40 and 80 days for January (left: blue) and July (right: red). Squares indicate $CRPSS_{pers}$, triangles $CRPSS_{clim}$ and boxplots are for correlation. The diamonds indicate the reliability of CRPS (on the same scale as CRPSS). Triangles are identical for all days, January and July. The boxplots for the correlation indicate the spread across the 100 member ensemble forecasts.

One the one hand, the $CRPSS_{clim}$ values do not depend on the season (identical triangles in Fig. 5). On the other hand, $CRPSS_{pers}$ values are higher in July than January for lead times $T \leq 10$ days, and lower for lead times $T \geq 40$ days (squares in Fig. 5). This means that the climatology forecast tends to be better than the persistence forecast for $T > 5$ days (squares higher than triangles in Fig. 5), which can be anticipated because of the inherent lag of the persistence forecast.

5      The correlation scores decrease with lead time $T$. The correlation skill is higher in January than in July. It is no longer significantly positive for $T$ larger than 40 days (25 to 75th quantile intervals contain the 0 value). The correlation score values range between 0.65 and 0.82 for $T = 5$ day forecasts, and 0.45 and 0.77 for $T = 10$ day forecasts, depending on the season. This is consistent with the NAO forecast of the Climate Prediction Center ($r = 0.69$ for a 10 day forecast). The correlation score is still significantly positive for $T = 20$ days. The higher correlation scores over the whole year (not shown) reflect a (small)

10    seasonal cycle of the NAO index. This artificially enhances the score for those lead times because SLP analogue predictands tend to reproduce the seasonality of the SLP field (by construction of the simulation procedure), and the NAO index and SLP variations are closely linked on monthly time scales (by construction of the NAO index).

     For comparison purposes, the multivariate autoregressive model NAO time series are shown in Fig. 6. The skill scores (correlation and CRPSS) for the NAO index give positive values, but not as high as for the analogue forecast. Since this

weather generator is designed to yield stationary statistical properties, the score values do not depend on the season. CRPSS values for climatology range from 0.36 ($T = 5$ days) to 0.23 ($T = 80$ days). Those values are lower than for the analogue system for lead times lower than 20 days (Fig. 5, triangles). The correlation values decrease from 0.58 ($T = 5$ days) to 0.05 ($T = 80$ days), which is lower than for the analogue system (Fig. 5, boxplots).

## 4.2 European temperatures

The correlation and CRPSS values for daily mean temperature (TG) forecast are shown in Fig. 7. The values of CRPSS$_{\mathrm{pers}}$ (for a persistence reference) increases with lead time $T$. This is not surprising because the forecast for the next $T$ days is based on the average of the past $T$ days. Therefore, the persistence forecast is always "late" due to the strong seasonality of temperature variations.

The CRPSS$_{\mathrm{clim}}$ values decrease with $T$ and plateau near $\approx 0.2$. This skill score is still positive (albeit small) for a seasonal forecast. This positive average skill (CRPSS $> 0.2$) illustrates that the stochastic weather generator follows the seasonality of temperature variations. We note that the CRPSS values for temperature are higher than for the NAO index. This is explained by the seasonality of temperatures, which is more pronounced than in the daily NAO index.

The CRPSS values are rather consistent for the four of the stations (Toulouse, De Bilt, Berlin and Orly). The stochastic model CRPSS fares slightly worse at Madrid station.

The CRPS reliability values are shown in Fig. 7. Their absolute values are larger than for the NAO index, and need to be normalized by the variance of temperature (or the CRPS value itself), as the units of TG are tenths of degrees. The average relative reliability values for lead times lower than 10 days are also similar to what is reported by Hersbach (2000). The reliability values seem to decrease with lead times in winter. They peak at lead times of 40 days in the summer (except for Berlin, where the peak is at 20 days in the summer), then decrease.

The correlation scores for January and July decrease with lead time $T$. The correlation score values for all days are above 0.97 due to the seasonality of temperatures and forecasts. Since this is not informative, this is not shown in Fig. 7. The correlations are always significantly positive for Toulouse, De Bilt, Berlin and Orly. The summer correlation intervals contain the 0 value at Madrid. This is probably due to the fact that temperature is not linked to the atmospheric circulation in the summer, but rather to local processes of evapo-transpiration (Schaer et al., 1999; Seneviratne et al., 2006). The distribution of the correlation scores (boxplots in Fig. 7) is significantly positive for lead times up to 20 days. It becomes stable near values of 0.2 (or increases) for lead times larger than 40 days. This indicates that there is certainly an artificial predictability beyond those lead times, that show an upper limit of forecasts for this system.

The mAR1 system for temperature is not designed to yield a seasonal cycle (contrary to the analogue system). Therefore, the skill scores of this system for temperatures are negative (for CRPSS) or with non significant correlations.

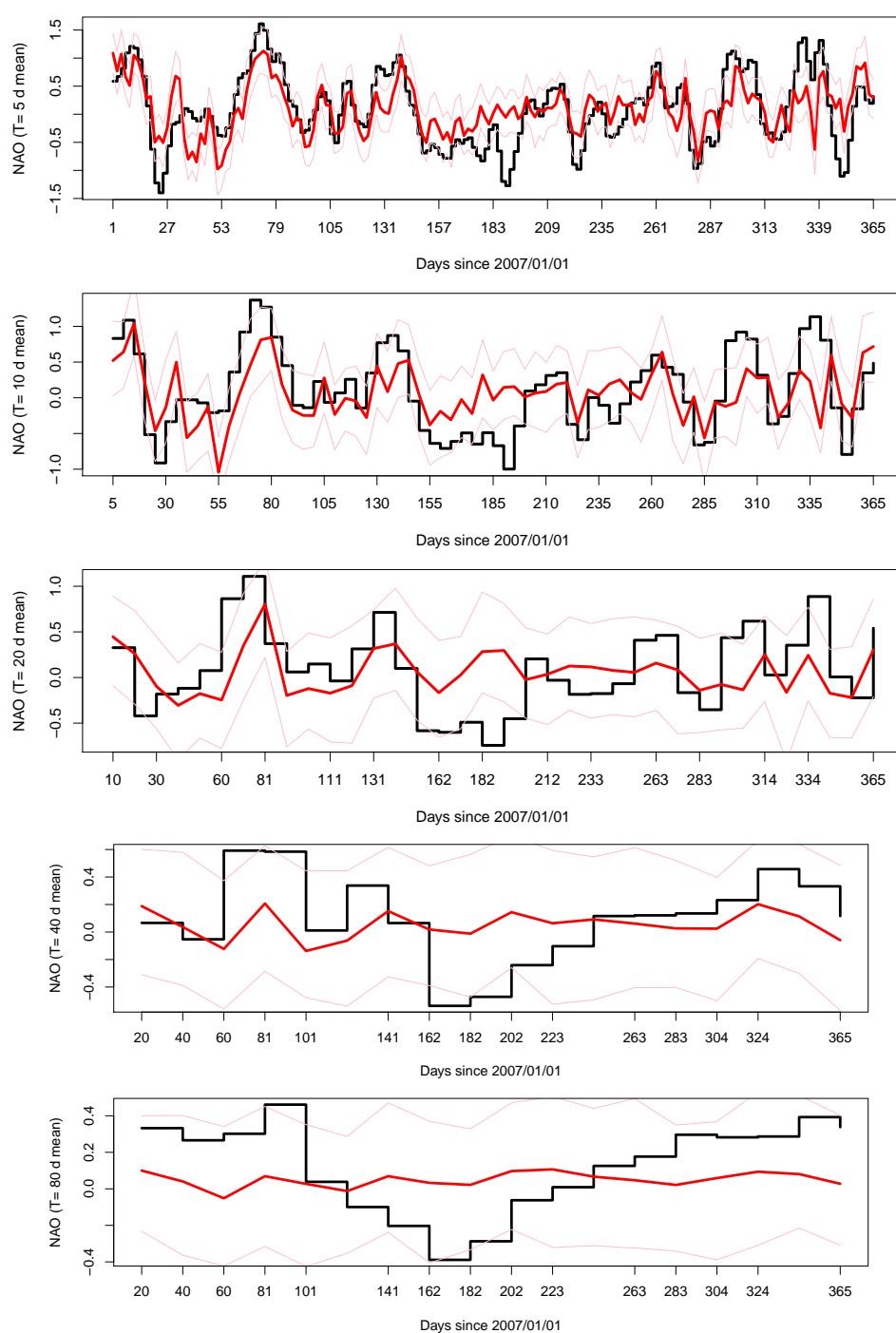

**Figure 6.** Multivariate autoregressive (mAR1) model time series of ensemble forecasts for 2007, for lead times $T \in \{5, 10, 20, 40, 80\}$ days. Black lines represent observed averages over lead times $T$. Red lines represent the median of 100 simulations; pink lines represent the 5th and 95th quantiles of the 100 member ensemble.

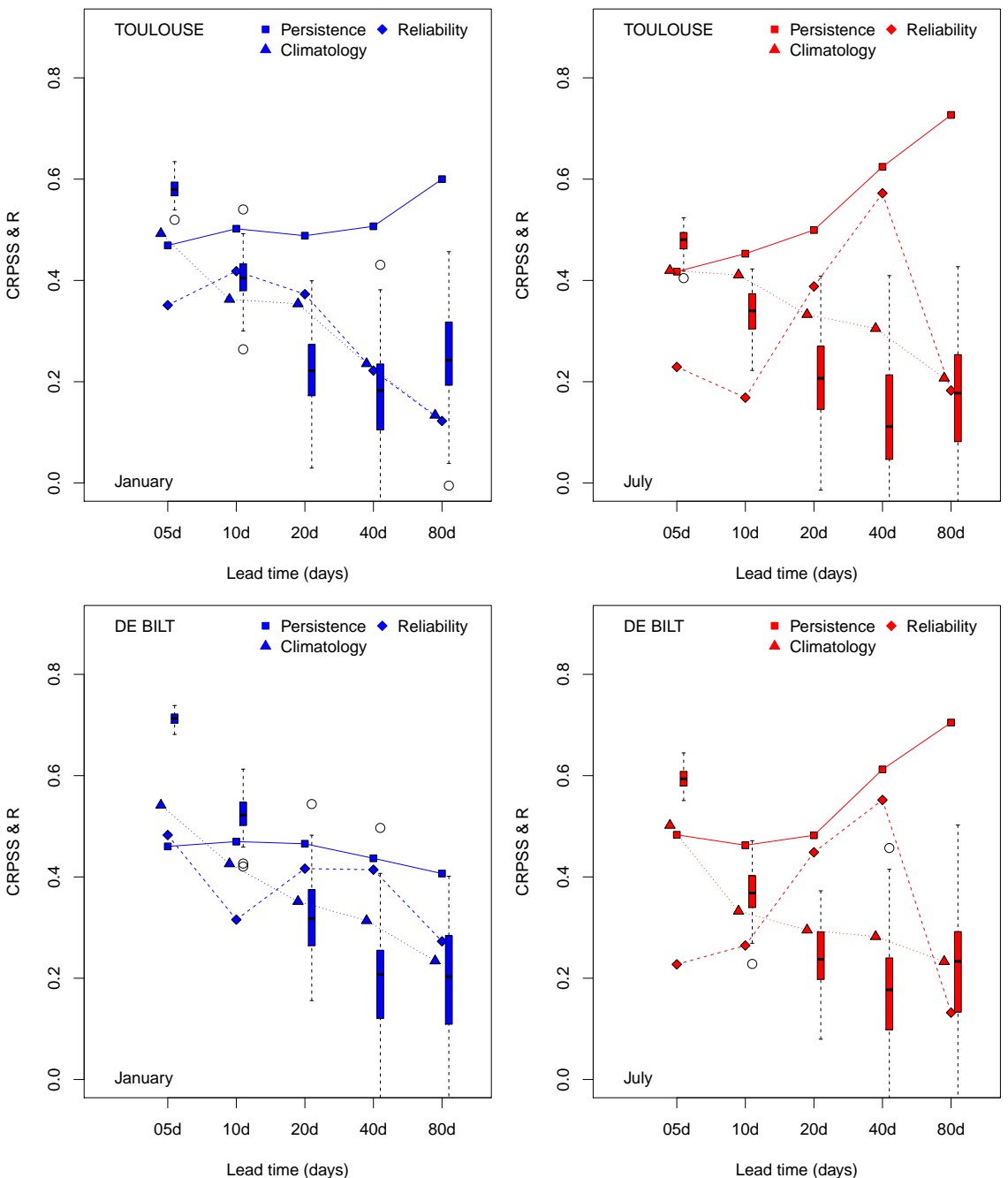

**Figure 7.** Skill scores for mean daily temperature in Toulouse, De Bilt, for lead times $T$ of 5, 10, 20, 40 and 80 days. Square indicate $CRPSS_{pers}$, triangles $CRPSS_{clim}$ and circles are fore correlation. The diamonds indicate the reliability of CRPS (on the same scale as CRPSS). Blue symbols (left) are for January and red symbols (right) are for July. Triangles are identical for January and July. The boxplots for the correlation indicate the spread across the 100 member ensemble forecasts.

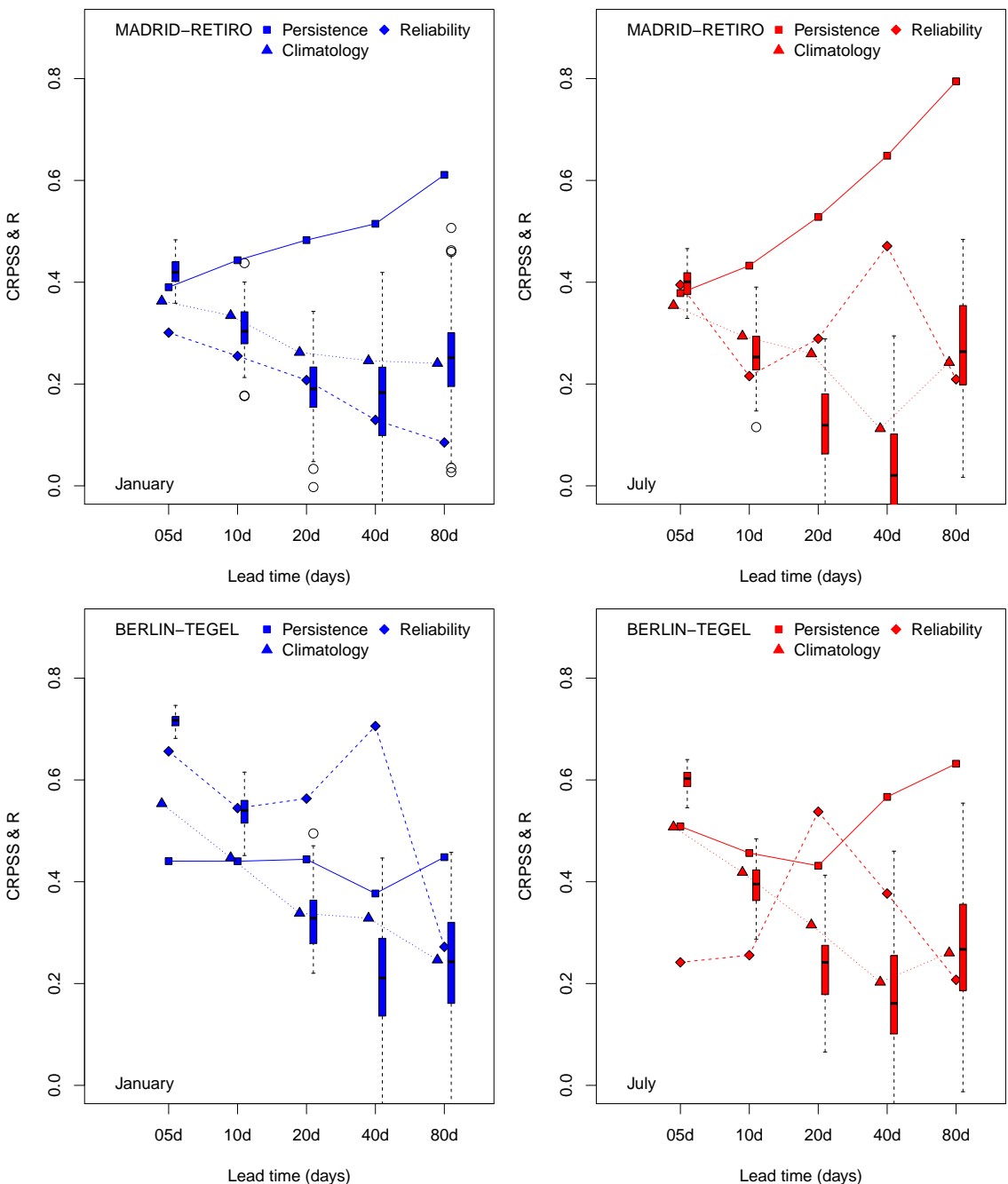

**Figure 7.** Skill scores for temperature in Madrid and Berlin (continued).

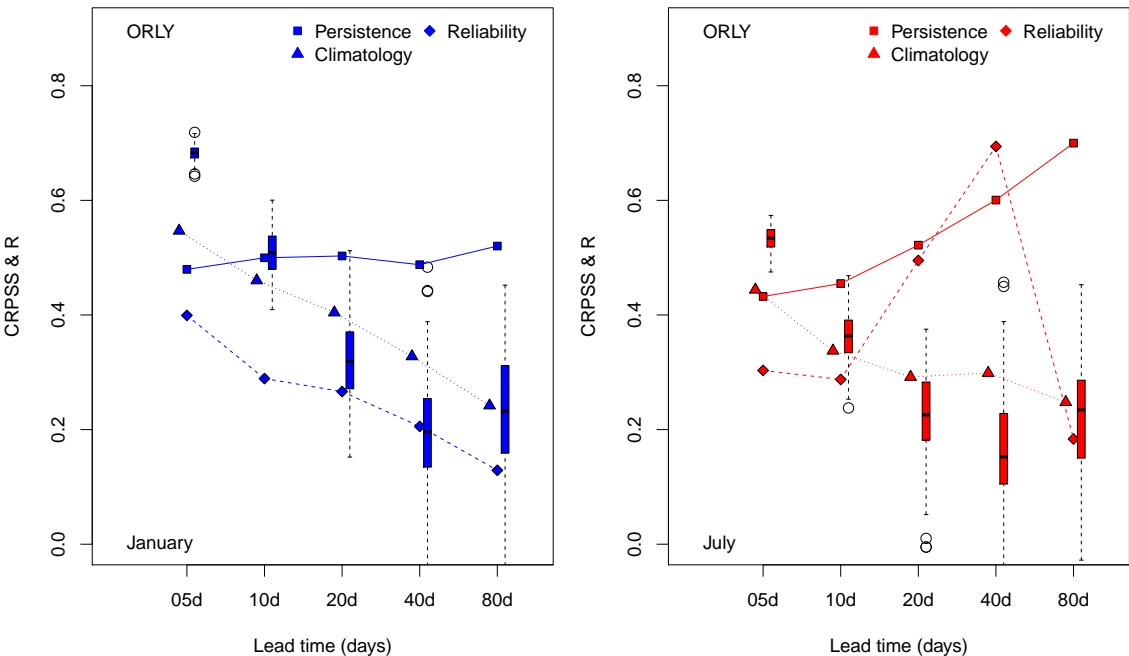

**Figure 7.** Skill scores for temperature in Orly (continued).

## 5 Conclusions

We have presented a system to generate ensembles of stochastic simulations of the atmospheric circulation, based on pre-computed analogues of circulation. This system is fairly light in terms of computing resources as it can be run on a (reasonably powerful) personal computer. The most fundamental assumption of the system is that the variable to be predicted is linked to
the atmospheric circulation. The geographical window for the computation of analogues needs to be adjusted to the variable to be predicted, so that a prior expertise is necessary for this analogue forecast system. This implies that this approach would not be adequate for variables that are not connected in any way to the atmospheric circulation (here approximated by SLP). The use of other atmospheric fields (e.g. geopotential heights) might increase the skill of the system. The computation of analogues with other parameters (geographical zone, atmospheric predictor, type of reanalysis, climate model output, etc.) can be easily
performed with a web processing service (Hempelmann et al., 2018).

We have tested the performance of the system to simulate an NAO index and temperature variations in five European stations. The performance of such a system cannot beat a meteorological or seasonal forecast with a full-scale atmospheric model (Scaife et al., 2014), but its skill is positive, even at a monthly time scale, with a rather modest computational cost. From the combination of several skill scores (from CRPS and correlation), we obtain a forecast limit of 40 days, beyond which the
interpretation of score values is artificial. We emphasize that the forecast is done on averages over lead times, not on the last value of the lead time.

The reason for the positive skill (especially against climatology) remains to be elucidated, especially for lead times longer than 20 days. We conjecture that the information contained in the initial condition (as done with regular weather forecasts) actually controls the mean behavior of the trajectories from that initial condition. But such a skill is actually "concentrated" in the first few days, because the trajectories tend to converge to the climatology after 20 days. The combination of several skill scores shows that such a system is not appropriate for ensemble forecasts beyond lead times of 40 days, which is lower than what is reported by Baker et al. (2018) for a meteorological forecast of the NAO.

Although the forecast system is random, it contains elements of the dynamics of the atmosphere, from the choice of the analogues. This system is consistently better than a simple multivariate autoregressive (mAR1) model for lead times shorter than 20 days. Since the seasonal cycle is naturally embedded in the analogues simulations, there is no need to parameterize it, contrary to the mAR1 model.

Recent experimental results in chaotic systems have shown that a well tuned neural network algorithm could simulate efficiently the trajectories of a chaotic dynamical system (Pathak et al., 2018b). Our system is an extreme simplification of an artificial intelligence algorithm, but it does demonstrate the forecast skill of such approaches. The advantage here is the physical constraint between the atmospheric circulation and the variables to be simulated.

This system was tested on temperature for five European datasets. This could be extended to precipitation or wind speed. If a real-time forecast is to be performed, we emphasize that only the predictor (here, SLP) needs to be regularly updated for the computation of analogues.

The goal of such a system is not to replace ensemble numerical weather/seasonal forecast. Rather, it can refine the usual references (climatology and persistence) for the evaluation of skill scores. This would create a third "machine learning" reference for CRPSS that might be harder to beat than the classical references.

*Code and data availability.* The code for the computation of analogues is available at (free CeCILL license): https://a2c2.lsce.ipsl.fr/index. php/deliverables/101-analogue-software and at: https://github.com/bird-house/blackswan

The code for simulations is available at: https://a2c2.lsce.ipsl.fr/index.php/deliverables under a free CeCILL licence (http://www.cecill. info/licences.fr.html).

The temperature data are available at: https://www.ecad.eu

The NAO index data are available at: http://www.cpc.ncep.noaa.gov/products/precip/CWlink/pna/nao.shtml.

The NCEP reanalysis SLP data is available at: https://www.esrl.noaa.gov/psd/data/gridded/data.ncep.reanalysis.html

*Author contributions.* PY wrote the codes, designed the experiments. CD participated to the writing of the manuscript.

*Competing interests.* The authors declare no competing interest.

*Disclaimer.* TEXT

*Acknowledgements.* This work was supported by a grant from the Labex-IPSL and ERC grant No. 338965-A2C2. We thank Mariette Lamige and Zhongya Liu who performed preliminary analyses during their training periods. We thank the two anonymous reviewers for their suggestions to use the CRPS decomposition and a simple parametric stochastic model.

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
