# Peer review of "Stochastic Ensemble Climate Forecast with an Analogue Model"

_Geoscientific Model Development, 2018_

## Referee Comment (RC1) · Anonymous Referee #1 · 16 Oct 2018

General comments:

In this paper, authors use a low-cost stochastic analogue forecasting method to predict the NAO index and ground temperatures in specific locations. The idea is the following: find 20 analog situations using the sea level pressure at time t, randomly choose 1 of the 20 analogs (using a proper distance), take the corresponding successor to make the prediction at t+1, apply the same procedure until lead time t+T. Authors repeat this statistical forecast and obtain a stochastic ensemble forecast of 100 simulated trajectories. The method is original and have good performance compared to classic ones, using persistence or climatology. The introduction is very clear and is a good summary of stochastic weather generators and analog methods. However, quality of

the figures needs to be improved.

Specific comments:

- The stochastic analog forecast presented here is a nonparametric approach (in a statistical sense). At some points, the reader would like to have a comparison with simple parametric methods like an autoregressive model, building a linear regression between the SLP at time t and NAO index or ground temperature at time t+1. Another option is to build a local linear regression between the 20 analogs and 20 successors. In that case, the biases given highlighted in the q-q plots should be reduced and quality of the prediction should be improved. But the use of low-rank methods (like Partial Least Squares method) must be used. Note that using such parametric methods can also lead to stochastic forecasts, when randomly sampling on the distribution function (e.g., Gaussian with the estimated mean and covariance) of the successors.

- The quality of the figures needs to be significantly improved:
  - Fig. 1, can you remove the 2nd map an put only the 5 points of interest in the 1st map?
  - Fig. 2, what do you mean by observed average. Is it really useful? Where are the median analog forecasts? Please use T instead of N in the legend.
  - Fig. 3, plot only 1 legend (for instance in the bottom left sub-figure)? Be careful with the y-label on the right sub-figures.
  - Fig. 4-5, authors should separate Jan and Jul in 2 sub-figures (not necessarily to plot "all"). Please connect the [squares, dots, triangles] between different lead times. Use a classic boxplot to represent error bars.

Technical corrections:

- Avoid the use of "dynamical" and use "dynamic" instead.

- Can you explain the difference between "predictand" and "predictor"? Avoid the use of predictand?

- Can you remind the difference between positive and negative values of the NAO index?
* * *

---

## Referee Comment (RC2) · Anonymous Referee #2 · 19 Oct 2018

In this manuscript the authors develop an ensemble forecasting system using an analog-based weather generator. They test this ensemble forecasting system for NAO and the temperature at several weather stations. They focus on the forecasts of temporal averages from 5 to 80 days. The forecast is made for each averaging period at the first corresponding lead time. The skill of these forecasts are evaluated through skill scores (the correlation and the continuous rank probability score, CRPS, the latter being well adapted to ensemble forecasts). The authors claim that there is some skill of the temperature and NAO up to seasonal time scales. I am not convinced by the system they propose, nor by the skill they found, for three important reasons:

(i) The system they propose suffers from a very important drawback, which is the progressive convergence toward the climatological mean as illustrated on the right column

of Figure 3. There is only little variability of the forecasts for long time averages, indicating that the ensemble forecast is unreliable. This makes of this system a very poor probabilistic forecasting system, since the forecasts do not span the set of possible values of the observed variable. Reliability is one essential ingredient of ensemble forecasts that can also be easily checked with the decomposition of the CRPS in reliability and resolution. I therefore do not consider this ensemble system appropriate.

(ii) It is not clear at all to me why the authors are looking at the first lead time of the 5, ... 80 days averages. Using this approach, one can certainly expect that if one start from an initial state close to the reality, the forecast of the averages will always be better than the climatological average (provided we have access to an infinite sample). In other words some positive correlation will always be present, even if it is very small. This skill is artificial (due to averaging from the initial state) and I am wondering why the authors did not have looked at the skill of the daily values of NAO or temperatures. My guess is that there is no skill beyond a month or so.

(iii) The analysis of the skill of ensemble forecasts should be done with appropriate tools. The CRPSS is one of them, but it is much more important to look at its decomposition in reliability, resolution and uncertainty. These are standard tools that can be found in classical books or papers (e.g. H. Hersbach, 2000, Weather and Forecasting, 15, 559-570).

Some additional (less important) points

1. The algorithm of page 4 (section 3.2) is far from clear. It would be nice to visualize the algorithm, together with the relations that are used for evaluating the weights.

2. Page 6, line 5. Is S=N? This is not clear to me.

3. An additional concern I have is the comparison with the persistence in Figs 4 and 5. It seems to me that the observables based on persistence display a higher variability than the forecasts constructed here (that are converging to the climatology). I there-

fore suspect that the reliability of the persistent forecast is better than the one of the stochastic forecasts (the reliability term in the CRPS decomposition should be smaller for the persistence case), which is not reflected here in the analysis of the CRPSS. I would be very useful to evaluate the different terms of the CRPS to clarify the difference between the two systems. This will allow in particular to clarify why one gets 0.45 for all averages for NAO and why the skill increases for temperature.

Based on these considerations, I do not recommend publication of this manuscript.

---

## Author Comment (AC1) · 14 Dec 2018

The replies to referees are indicated in red. We thank the reviewers for pointing out unclear points in the manuscript.

**Referee#1**

In this paper, authors use a low-cost stochastic analogue forecasting method to predict the NAO index and ground temperatures in specific locations. The idea is the following: find 20 analog situations using the sea level pressure at time t, randomly choose 1 of the 20 analogs (using a proper distance), take the corresponding successor to make the prediction at t+1, apply the same procedure until lead time t+T. Authors repeat this statistical forecast and obtain a stochastic ensemble forecast of 100 simulated trajectories. The method is original and have good performance compared to classic ones, using persistence or climatology. The introduction is very clear and is a good summary of stochastic weather generators and analog methods. However, quality of the figures needs to be improved.

**Specific comments:**

• The stochastic analog forecast presented here is a nonparametric approach (in a statistical sense). At some points, the reader would like to have a comparison with simple parametric methods like an autoregressive model, building a linear regression between the SLP at time t and NAO index or ground temperature at time t+1. Another option is to build a local linear regression between the 20 analogs and 20 successors. In that case, the biases given highlighted in the q-q plots should be reduced and quality of the prediction should be improved. But the use of low-rank methods (like Partial Least Squares method) must be used. Note that using such parametric methods can also lead to stochastic forecasts, when randomly sampling on the distribution function (e.g., Gaussian with the estimated mean and covariance) of the successors.

This is an interesting suggestion, albeit quite unusual (with respect to the available literature). We built a multivariate autoregressive (mAR1) model on SLP. To simplify numerical problems, the mAR1 model is done on the first ten principal components of North Atlantic SLP (representing approx. 80% of the variance):

$$R_{t+1} = AR_t + B_t,$$

where $R_t$ is a vector of 10 PCs, $A$ is a $10 \times 10$ "persistence" matrix, and $B_t$ is a 10-variate Gaussian centered white noise with covariance matrix $\Sigma$.

The mAR1 coefficients $A$ and $\Sigma$ are determined from the covariance $C(0)$ and lag-1 covariance $C(1)$ matrices of SLP:

$$A = C(1)^t C(0)^{-1},$$
$$\Sigma = C(0) - C(1)^t A.$$

This procedure is similar to what was done by Michelangeli et al. (J. Atmos. Sci. 1995) to simulate a multivariate AR1 process that mimics atmospheric geopotential heights.

Ensembles of mAR1 simulations can be performed, with initial conditions from observed values of SLP, at incremental times. This is similar to the analogue weather generator of SLP presented in the paper.

We performed a multilinear regression between the five temperature series and NAO index and the preceding values of SLP principal components:

$$X_t = [T_t^1, \dots, T_t^5, NAO_t] = aSLP_{t-1} + b + e_t.$$

This multivariate regression is applied to the mAR1 model to perform ensembles of forecasts of temperatures and NAO index. For each realization, averages over lead times between 5 and 80 days are then performed. Such a simple model cannot reproduce a seasonal cycle of temperature (unless it is explicitly added, which we did not do). Therefore, only comparisons on the warmest (July) or coldest (January) months would be

meaningful. Such a problem does not occur with the NAO index, which does not yield a clear seasonality.

We computed the CRPSS and correlation of this stochastic model (mAR1). The skill scores always give negative (or non significantly positive) values with respect to references for temperature or NAO index. Therefore, an autoregressive stochastic model does not provide improvements over references.

In addition (we had not mentioned it but we will in the revised text), a stochastic IID perturbation is always added to the reference (climatological, persistence) forecasts. This is necessary because we compare probability distributions. This is an even simpler first order parametric stochastic model.

This will be discussed in the manuscript.

• The quality of the figures needs to be significantly improved:
−Fig. 1, can you remove the 2nd map and put only the 5 points of interest in the 1st map?
OK. The 2nd panel was removed and the 5 stations were added on the bigger map.

[Figure]

−Fig. 2, what do you mean by observed average. Is it really useful? Where are the median analog forecasts? Please use T instead of N in the legend.
It should have been "the average of observed temperatures TG between Jan. 1st 2007 to the lead time T". This is the values that we try to forecast. The legend of the figure is changed (T rather than N). Thank you for pointing this out.

[Figure]

−Fig. 3, plot only 1 legend (for instance in the bottom left sub-figure)? Be careful with the y-label on the right sub-figures.
OK. The legends were removed, and grouped in an additional panel.

−Fig. 4-5, authors should separate Jan and Jul in 2 sub-figures (not necessarily to plot "all"). Please connect the [squares, dots, triangles] between different lead times. Use a classic boxplot to represent error bars.
OK. The figures are splitted in two panels. The error bars represented the 95% confidence interval obtained from a usual formula on uncertainty on the correlation (see H. von Storch and F. Zwiers, Statistical Analysis in Climate Research, 1999, Cambridge University Press, sec. 8.2.3) between the median of forecasts and observed values. The new figures now represent the spread of correlations between realization members and observations with boxplots. The interpretation of confidence intervals is hence different (but the mean values are the same).

**Technical corrections:**

• Avoid the use of "dynamical" and use "dynamic" instead.

We keep the adjective "dynamical" when referring to "dynamical systems". This is how it is used in textbooks, journal names, etc. The adjective was changed to "dynamic" when referring to the simulation mode of the stochastic weather generator.

• Can you explain the difference between "predictand" and "predictor"? Avoid the use of predictand?

Predictand is the variable that we want to predict. Predictor is the variable that is used to predict the predictand. There was a confusion p. 12, l. 32, which is now corrected.

• Can you remind the difference between positive and negative values of the NAO index?

A sentence is added to explain the pressure features during high and low values of the NAO index (p. 2, near l. 30).

---

## Author Comment (AC2) · 14 Dec 2018

The replies to referees are indicated in red. We thank the reviewers for pointing out unclear points in the manuscript.

**Referee#2**

In this manuscript the authors develop an ensemble forecasting system using an analog-based weather generator. They test this ensemble forecasting system for NAO and the temperature at several weather stations. They focus on the forecasts of temporal averages from 5 to 80 days. The forecast is made for each averaging period at the first corresponding lead time. The skill of these forecasts are evaluated through skill scores (the correlation and the continuous rank probability score, CRPS, the latter being well adapted to ensemble forecasts). The authors claim that there is some skill of the temperature and NAO up to seasonal time scales. I am not convinced by the system they propose, nor by the skill they found, for three important reasons:

(i) The system they propose suffers from a very important drawback, which is the progressive convergence toward the climatological mean as illustrated on the right column of Figure 3. There is only little variability of the forecasts for long time averages, indicating that the ensemble forecast is unreliable. This makes of this system a very poor probabilistic forecasting system, since the forecasts do not span the set of possible values of the observed variable. Reliability is one essential ingredient of ensemble forecasts that can also be easily checked with the decomposition of the CRPS in reliability and resolution. I therefore do not consider this ensemble system appropriate.

We never hide the fact that there is a convergence towards climatology (this is mentioned in the text). But long term forecasts with full scale climate models yield the same feature (as outlined by Hersbach (2000) and others). Our claim is that this system does a better job than usual references (Climatology or Persistence) or AR1 models (see response to referee #1). The ease of use of this system makes it possible at low cost to investigate the limit for large lead times. We consider that the fact that the scores are positive for shorter lead times (20 days ahead) is interesting. We now mention (and use) the CRPS decomposition of Hersbach (2000) in terms of reliability and potential CRPS.

(ii) It is not clear at all to me why the authors are looking at the first lead time of the 5, … 80 days averages. Using this approach, one can certainly expect that if one start from an initial state close to the reality, the forecast of the averages will always be better than the climatological average (provided we have access to an infinite sample). In other words some positive correlation will always be present, even if it is very small. This skill is artificial (due to averaging from the initial state) and I am wondering why the authors did not have looked at the skill of the daily values of NAO or temperatures. My guess is that there is no skill beyond a month or so.

We never claim the contrary and discussed it in the text. Starting from an observed state, daily trajectories tend to diverge from each other. The computing T-averages for various lead times allows accessing to the limit of predictability of our system.
But if an autoregressive model (mAR1, see response to referee#1) is initialized from observations, there is NO skill (correlation or CRPS). As stated in the text, we do not consider that the system has any skill beyond a month.

(iii) The analysis of the skill of ensemble forecasts should be done with appropriate tools. The CRPSS is one of them, but it is much more important to look at its decomposition in reliability, resolution and uncertainty. These are standard tools that can be found in classical books or papers (e.g. H. Hersbach, 2000, Weather and Forecasting, 15, 559-570).

Thank you for this suggestion. We add a discussion on the decomposition of CRPS (citing the paper of Hersbach 2000) in terms of reliability and potential CRPS. The problem with reliability is that its magnitude depends on the unit of the variable to be predicted (as discussed by Hersbach 2000). The results reported by Hersbach give very small values of reliability (for precipitation forecast) when the ECMWF analysis is used. But those numbers are small because the variable values to predicted are small.
We used the R package "verification" (by E. Gilleland) to compute this decomposition. The relative of variations of reliability that we obtain for temperature or NAO forecasts is in the same range of what is reported in the paper of Hersbach (2000) for lead times of 5 to 10 days. We now discuss the values of reliability, which appears in Figures 4-5. The reliability values for NAO are small ($\approx$ 8 10$^{-3}$), and the ratio to the CRPS value is in the same range of what is reported in Hersbach's paper.

**Some additional (less important) points**

1. The algorithm of page 4 (section 3.2) is far from clear. It would be nice to visualize the algorithm, together with the relations that are used for evaluating the weights.
OK. A graphical illustration is added (see below) to visualize the iteration procedure and the choice of weights to sample analogues.

[Figure]

*Illustration 1 : Schematic of the stochastic analogue weather generator. ymd indicates when absolute time is considered. md indicates when calendar time (i.e. time in the year) is used.*

2. Page 6, line 5. Is S=N? This is not clear to me.
We compute the N=20 best analogues for each day. At each time increment, we simulate S=100 trajectories, sampled from those 20 best daily analogues. For a lead time of, say, 10 days, there are 20$^{10}$ possible trajectories, which is far larger than S. This will be emphasized in the text.

3. An additional concern I have is the comparison with the persistence in Figs 4 and 5. It seems to me that the observables based on persistence display a higher variability than the forecasts constructed here (that are converging to the climatology). I therefore suspect

that the reliability of the persistent forecast is better than the one of the stochastic forecasts (the reliability term in the CRPS decomposition should be smaller for the persistence case), which is not reflected here in the analysis of the CRPSS. I would be very useful to evaluate the different terms of the CRPS to clarify the difference between the two systems. This will allow in particular to clarify why one gets 0.45 for all averages for NAO and why the skill increases for temperature.

A discussion on the CRPS decomposition for the different forecasts is added. The reliability value of CRPS for the persistence or the climatology give higher values (roughly twice larger) than for our model.